# The Narrative Bodies of James Baldwin: A Discussion of Literary and Sartorial Style

Sha'Mira Covington [1,2]

1 Institute for African American Studies, University of Georgia, Athens, GA 30609, USA; scovington@uga.edu
2 Department of Textiles, Merchandising, and Interiors, University of Georgia, Athens, GA 30609, USA

**Abstract:** Inspired by Terry Newman's literary and sartorial analysis of writers in her book *Legendary Authors and the Clothes They Wore*, I analyze James Baldwin's literary and sartorial style using excerpts from his works and archival photography. I also add a signifier/signified analysis using social semiotic theory. According to De Saussure, there are two main parts to any sign, the signifier, which connotes any material *thing*, and the signified, which is the meaning that is made of that *thing* by the receiver. Social semiotics changes the focus from the sign to the way people use semiotic resources to produce communicative artifacts, collectively. In the semiotic tradition, I extend the literary text (*Go Tell it on the Mountain*, *Another Country*, and *Just Above My Head*) to a larger reading of the culture in which it was created and to the more universal structures that are inherent within it. Clothing is also considered a critical semiotic resource because it is viewed as a sign that signifies a particular meaning. In my analysis, I illuminate how Baldwin's sartorial style is a mirror (signifier) to reflect his literary style and reflects the creative and spiritual (signified) essence of his work, connected to and with collective Black narratives of style.

**Keywords:** James Baldwin; literary style; sartorial style; fashion; semiotics





## 1. Introduction

Fashion is a language. It can serve as a wordless means to communicate a prevailing message to the world around us. It, in turn, becomes our language which speaks to a certain story about us. The stories that clothing communicate without words are irrefutable. This process can be more sophisticatedly understood through social semiotics. Generally, semiotics is the study of sign processes and systems in culture. It is an important aspect of understanding media, such as photography, film, and advertisements (Nöth 1997). It is particularly useful for analyzing signs as vehicles to propagate meaning. According to Saussure, there are two main parts to any sign: the signifier, which connotes any material *thing*, and the signified, which is the semiotic resource, viewed as a sign that signifies a particular meaning (De Saussure 2001). For example, a firefighter uniform (sign) is a specific type of clothing that associates a person with a specific profession as a first responder. The uniform is symbolic because it represents authority, respect, and safety (signifier) based on collective social agreements and norms.

Clothing rarely operates in isolation as a semiotic resource; therefore, in social semiotics, the focus changes from the sign to the way people use semiotic resources, like clothing, to produce communicative artifacts and events as well as interpret them in the context of collective situations and practices (Van Leeuwen 2005). Van Leeuwen's framework can be particularly useful for understanding how fashion can tell a story. As Newman (2017) suggests in *Legendary Authors and the Clothes They Wore*, the sartorial choices that writers make are deeply connected to the literary narrative choices that they make (Newman 2017). Moreover, if fashion can tell a story, which it does, a writer's sartorial choices might reflect the creative and spiritual essence of their work.

When referring to the sartorial, as Tulloch (2016) in *The Birth of Cool: Style Narratives of the African Diaspora*, I use "style" to mean the agency in the construction of self through

the assemblage of garments and accessories. Tulloch also contends that "in the study of Black people and the African diaspora, the concept of style encompasses myriad routes and connections, flows and tensions that originate from the analytical of Africa and its diaspora" (p. 5). Therefore, my analysis hinges on the concept that James Baldwin's sartorial choices not only represent his clothing, but broader collective Black narratives around themes such as spirituality, belonging, kinship, liberation, and forgiveness.

I use the semiotic frame because fashion is considered a system of signification, which was first introduced by Roland Barthes (Barthes et al. 1983) and has been a dominant frame since the development of the fashion studies discipline. From semiotic perspectives, fashion is a collective institution, a system of values and mode of communication which is produced out of the material world. Clothing semiotics is also intimately entwined with body semiotics, gestures, and facial expressions in the creation and communication of meaning. Using a semiotic reading, I ask the question: Are there connections between Baldwin's individual literary and sartorial style to collective Black narratives of style?

James Baldwin is a model study for exploring the signature sartorial and literary style of an author, particularly at this moment of political and social circumstance. His ideas and his image are as relevant and timely in 2022 as they were during the height of his activities. Like many Black sartorial and creative choices today, Baldwin's social and political sentiments were evident in his image. His voice, writing, and fashion have left a lasting impression. My aim is to make links through excerpts that exemplify Baldwin's original writing style in specific works and his sartorial style during the same decade. The works of interest are *Go Tell it on the Mountain* (1953), *Another Country* (1962), and *Just Above My Head* (1979). I chose these novels based on their cultural relevance, chronology, and literary development. I foreground each section with a long-standing analysis of his works to highlight how Baldwin's literary storytelling informs his sartorial storytelling.

I use archival photography as the main source of sartorial analysis. It is important to note that images can contribute to subtle, unconscious, and often nonverbal racist ideas based on historical visual representations of Black people. There is a dichotomy between how images of Black people are interpreted by viewers, typically white viewers, and how Black people self-represent and produce meaning in imagery. Willis (2009) in *Posing Beauty: African American Images from the 1890s to the Present*, posits that photography is personal and political and challenges the assumptions made about Black subjects. Black imagery, whether as subjects or creators, usually comments on politics, culture, family, and/or history from both internal and external points of view. The use of archival photography, in this study, does not attend to the critical narrativizing of Black subjectivity, but instead focuses on the potential meanings of James Baldwin's sartorial narrative in imagery.

Using a social semiotic framework, I analyze literary excerpts and photographs of Baldwin in his clothing to highlight that fashion is indeed a mirror to reflect our individual innermost emotions and ideas, as well as collective narratives.

## 2. Results: The Intersections of Baldwin's Literary and Sartorial Style

### 2.1. Go Tell It on the Mountain

Baldwin's *Go Tell it on the Mountain* (1953; Baldwin and Morrison 1998) is his first major work as an author. The book is semi-autobiographical and imitates the troubled relationship Baldwin had with his own stepfather. The storyline follows themes of strained relationships, guilt, racism, and religion. **The book's protagonist, John Grimes, resembles Baldwin himself.** Both are young, Black, Harlem-born, and have stepfathers who are abusive preachers. Baldwin's stepfather also thought Baldwin was physically ugly, which likely impacted his views on his own appearance (Leeming 1994). The book, however, moves back and forth through generational stories, first dwelling on John (the main character), then moving on to others such as Gabriel (John's stepfather), Deborah (John's first wife), Elizabeth (Gabriel's wife and John's mother), and Florence (Gabriel's sister), among others. The book flashes back in time to explore each character's struggles to reveal how religion,

race, and strained interpersonal relationships aggravate their lives, often to the point of tragedy.

Early biblical references in the book indicate the importance of Christianity in Baldwin's life. *Go Tell it on the Mountain* alludes to the Bible often, not just because the characters are connected with the church, but because Baldwin, too, had a deep connection with Black church culture. Like so many Black Christians, Baldwin knew the Bible intimately. During his high school years, Baldwin served as a youth minister in the Harlem Pentecostal church from the age of fourteen to seventeen, also preaching at a community church (NMAAHC n.d.).

His connection with the church and the Bible shows up often in *Go Tell it on the Mountain*. For example, in part one of the book, Baldwin references Romans 6:23 when comparing John's sin to the darkness of church on Saturday evenings. Baldwin writes " . . . It was like all this, and it was like the walls that witnessed and the placards on the walls which testified that the wages of sin was death" (p. 17). It is relevant, in light of the rest of the book, to reveal the full Bible verse which is, "For the wages of sin is death, but the gift of God is eternal life in Christ Jesus our Lord" (Romans 6:23). This quotation instigates two themes in the book. One is Gabriel's treatment of John as an illegitimate child. In Gabriel's mind, John is a sin. There are several instances that indicate that Gabriel resents John. The second theme is John's (and Baldwin's) gay identity perceived as sin. John struggles between his Christian upbringing and his sexual desires. Baldwin writes, " . . . he had sinned with his hands, a sin that was hard to forgive. In the school lavatory, alone, thinking of the boys, older, bigger, braver . . . " (p. 16). The biblical allusion within the storyline also reveals broader societal beliefs. For many Black people, the church historically acted as an anchor for the community, yet often condemned gay identity. The Black church and gay/queer identity have a complex relationship. While Black churches have historically condemned gay/queer identity, congregations are filled with gay parishioners. Baldwin's, and Black society's, racial and sexual politics serve as an example of the duplicity of material and spiritual dogma.

Baldwin uses biblical allusions liberally throughout the novel, and they have the effect of blurring the lines between material and spiritual life. For example, in a scene depicting Gabriel's hallucinatory church service experience, Baldwin writes, "On this threshing-floor the child was the soul that struggled to the light, and it was the church that was in labor, that did not cease to push and pull, calling in the name of Jesus . . . For the rebirth of the soul was perpetual; only rebirth every hour could stay the hand of Satan" (108). Baldwin uses the threshing floor allusion when explaining the altar floor of the church. The reader thus coincides the church altar with the Old Testament of the Bible which uses the image of the threshing floor to explain God's judgment. Rebirth, in this instance, refers to a spiritual rebirth. Redemption and salvation through rebirth can be understood in two ways: one, forgiveness for gay identity, if one was to adhere to heteronormative conformity, or two, the forgiveness of the Black church for condemning gay/queer identity.

Despite Black religious rhetoric on sexuality negatively impacting Baldwin's spiritual and religious life, his literary symbols were heavily inspired by his experiences in church and the Bible. His sartorial style was likely informed by the same space. His clothing choices during the same decade as *Go Tell it on the Mountain* also demonstrate his, and society's, keenness towards moral compliance. White European standards of beauty dominated the fashion world, during this time, and white European hair and facial characteristics were considered "normal" and desirable. Black people often tried to imitate those characteristics by straightening their hair and minimizing their Black features. Black American "mainstream" fashion of the 1950s was considered "Sunday's best," which meant that dressing up and looking one's best all the time, was expected. This manner of "put togetherness" seems to be common up until at least the early 1960s when young activists started to abandon "respectable" outfits for denim (Ford 2013).

Black Pentecostal experience is, in fact, essential to understanding both Baldwin's literary and sartorial style. In Figure 1, Baldwin is pictured in a black-and-white photograph

from 1955, taken by Carl Van Vechten, a white influential literary figure, cultural arbiter, and patron of the Harlem Renaissance. Van Vechten was also a queer man and part of the queer literary circle in New York (White 2014). In the photo, Baldwin is seated in front of a backdrop wearing a wide-collared, short-sleeved shirt. Baldwin's arms are crossed, likely how he was posed for the photograph, but also indicating an air of defensiveness or protection of his body. Although he is in his 30s in the photograph, his appearance is seemingly innocent and clean as a proper church boy might dress. Baldwin's elusive smile, in the image, suggests confidence, but also, amusement. Black fashion of the 1950s also exuded an essence of cool which Baldwin, although subtly, captures with his shirt unbuttoned at the neck. Additionally, during this time, and before it, the Harlem Renaissance crystallized Black queer culture in the mainstream (Schwarz 2003). Black, sexual, and queer liberation were happening simultaneously. In light of the time period, the photographer's, and Baldwin's queer identities, the image can also be read as a critique of the paradoxical intersections of Baldwin's oppressions and his freedoms, the Black church, and his queer social circle.

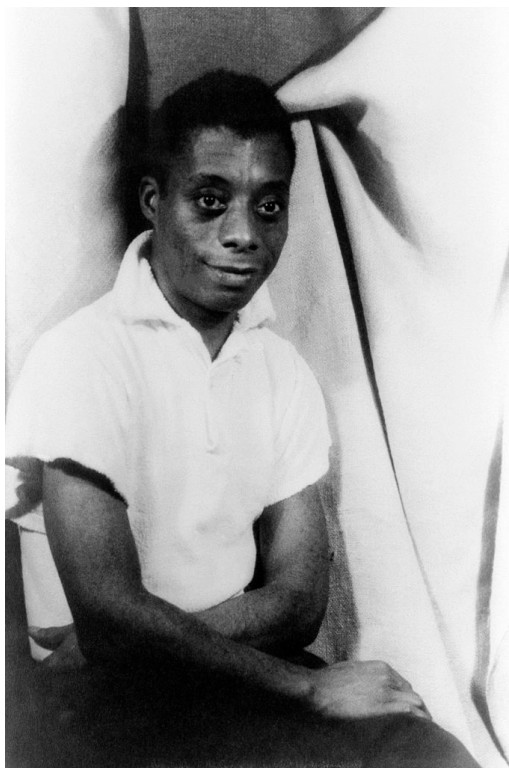

**Figure 1.** Photo by © CORBIS/Corbis via Getty Images.

By the 1950s, Harlem's Black dandy of the 1920s and 30s was a well-known persona. The Harlem Black dandy would have been considered a figure of urbanity and polished sophistication. Many attributed dandyism to sexuality and queer identity (Glick 2003), however, Black fashion followed some of the same sartorial codes. While grooming remained dapper and clean, the shirts, open suits, sports coats, and casual slacks of this time gave more of a relaxed and effortless look to anybody who donned the trends. The little details were what brought looks together like shirts with embroidery or hats paired with a shirt and pants rather than a suit. Emphasis was put on accessories and styling as well, such as watches, glasses, belts, or a popped collar as illustrated in Figure 1. During this time, Baldwin struggled with money and often had to borrow both money and clothing (Pavlic 2015). Despite this, he, in the historical tradition of Black fashion and sartorial culture, styled the clothes that he borrowed and could afford in unique ways to propagate particular meanings about himself and society.

For Black people, the church was seen as a safe place to experiment with sartorial self-definition (Covington and Medvedev 2019). The church played an important role in the lives of the Black community in providing social, political, and education support, all of which were evident through sartorial means. Additionally, white society had been conditioned to view Black people as threatening, which inspired Black churchgoers to dress as fine as possible to avoid negative white perceptions. Baldwin's use of biblical references in *Go Tell it on the Mountain* coincides with themes of racism in broader society. His clothing, too, seemingly agrees with these sentiments with additional layers of symbols to indicate his own subjectivities.

Like John Grimes in *Go Tell it on the Mountain*, Baldwin was tortured in the church and left at the age of seventeen. *Go Tell it on the Mountain* might reveal the multi-generational story of why he began to reject organized religion. Although Baldwin does not blatantly critique religion in the book, he does highlight the hypocrisy evident in it through Gabriel's character. He calls Gabriel the "anointed one"; however, he portrays him as a flawed character. In the book, Florence says, "[Gabriel] ain't got no right to be a preacher. He ain't no better'n nobody else" (p. 84). The negative portrayal of Gabriel, his story, and the connection to Baldwin's own stepfather has the effect of making Christianity appear hypocritical. In "*Letter from a Region in My Mind*" (1962), Baldwin writes about Christianity's promise and splendor, but is skeptical that it can be attained (Baldwin 1962).

Baldwin's continuing shift in religious thought might indicate his change in sartorial choices during this decade. Figure 2 is a William Cole (1953) photograph of Baldwin. Cole was also Baldwin's agent (Miller 2009). In the image, Baldwin is standing in front of a brick building wearing a wide collar over his suit lapel and donning a satin pocket square or handkerchief. His appearance is a bit more pensive and serious in this photo than in Figure 1. There is certainly a more evident essence of "Black cool" in his styling as he looks out into the distance. His groomed face and pinstripe suit, even if borrowed, indicate that he had some means of styling himself. Even the presence of his mustache alludes to asserting himself as a man or expressing a new persona. While James Baldwin had rejected mainstream religion, Black secular music began to heavily influence Black styling aided in solidifying the construction of "Black cool". The "Black cool" or "cool pose" style is prevalent in Black and urban groups. It includes flashy and provocative clothing, subtleties in body movements such as mannerisms, and gestures, and specific language that shows the dominant culture that one is strong and proud, despite racial status in American society (Majors and Billson 1992). In Figure 2, Baldwin exudes an air of confidence despite his struggle to find himself as a Black American writer. His sartorial choices follow collective Black narratives to assert an identity despite oppressive circumstances.

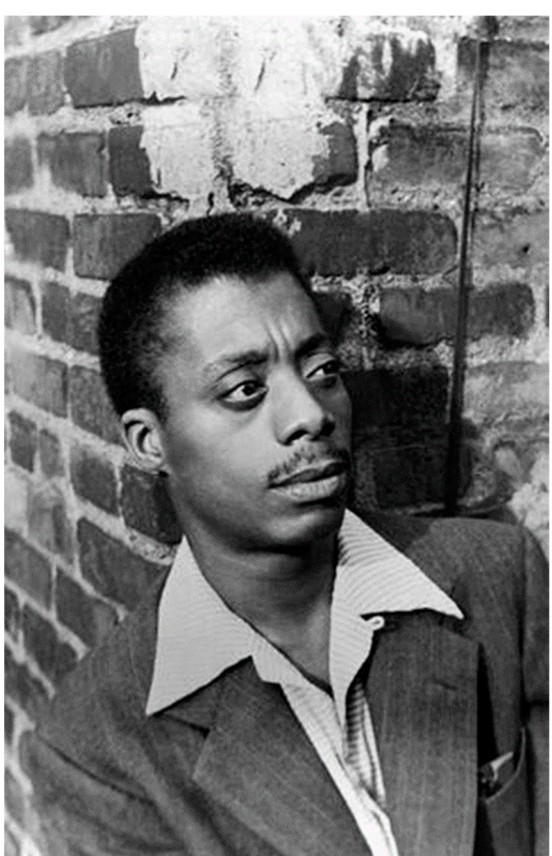

**Figure 2.** Courtesy: CSU Archives/Everett Collection via Alamy.

### 2.2. Another Country

*Another Country* (1962; 1998) centers on the death of Rufus Scott, a Black musician who lives in Greenwich Village. Rufus falls in love with a white woman, Leona, and struggles with himself and their interracial relationship. He also has several sexual relationships with men which allude to Baldwin's own sexuality as a gay man. After Rufus commits suicide, the book follows Rufus' friends Vivaldo, Cass and Richard Selenski, Eric, and Rufus' sister Ida. The major themes surround conflicts about racism, sexism, and homophobia. Baldwin seems to highlight the struggle of the individual against societal conformity through the characters. Additionally, Baldwin shows that conflict and hatred are often the result of individuals being "forced" to play into social roles that they do not fit into. He does this by highlighting the shared struggles of Black people, gay people, and poor people.

Shared loneliness is a reoccurring theme throughout *Another Country*, and it connects with his theme of secrecy. For example, "loneliness and contempt" drive Rufus to suicide. Loneliness is described as a disease, and the characters of *Another Country* are plagued with it. Baldwin writes about Vivaldo, "He had often thought of his loneliness, for example, as a condition which testified to his superiority. But people who were not superior were, nevertheless, extremely lonely—and unable to break out of their solitude precisely because they had no equipment with which to enter it" (p. 419). Vivaldo's fight with isolation creates a "strange climate of the city." He wonders where this "violent emptiness might drive an entire city" (p. 419). Perhaps the secrets that Baldwin wishes to unveil contribute to the loneliness that he writes about. It seems that Baldwin believes that revealing the secrets will remedy the loneliness. He writes, "One was continually being jostled, yet longed, at the same time, for the sense of others, for a human touch; and if one was never—it was the general complaint—left alone in New York, one had, still, to fight very hard in order not to perish of loneliness" (p. 570). The struggles of the individual characters in the book against societal conformity, whether it be through race, gender, or sexual orientation, are played

out through Baldwin's use of dichotomies and secrets. All of these issues were, of course, societal concerns too.

Music is also a theme that Baldwin uses to develop his characters and parallels with the cultural milieu in which *Another Country* was written. Rufus is a musician, and his sister Ida wants to be a singer. Vivaldo plays "James Pete Johnson and Bessie Smith . . . *Backwater Blues*" (p. 408). Baldwin even takes the time to give readers the experience of the song, writing, "*There's thousands of people*, Bessie now sang, *ain't got no place to go* . . . " (p. 408). Baldwin was a music lover. He used Black music as an influence and instigation. In his 1961 essay "The Discovery of What It Means to Be an American," he said that blues singer Bessie Smith's recordings helped him to write his first novel (Baldwin 1961). He also championed the work of jazz and blues artists in many other essays, works of fiction, and interviews (Lordi 2016).

Historically, music has been a means for Black people to express their history and present state. The blues—a style of music that evolved from southern Black secular songs in the colonial period—addresses themes like "work, love, death, lynching," which, for the Black population in the United States, can be summarized as "the Facts of Life" they have to cope with every day. As the Black Power Movement of the late 1960s began to flourish, music, politics, and dress were more apparently fused together. Baldwin's style during this time illustrated this shift.

In a photo of James Baldwin and James Meredith in 1963 (not pictured), Baldwin is wearing a slim-cut suit jacket, paired with plain black pants, a white dress shirt, and a skinny tie. Jazz musicians in the 60s wore a similar ensemble, donning slim suits in a range of fabrics, styles, and patterns. Besides their styling, musicians also used their voices for political messaging, and activists used popular songs as a part of their protest. For example, in 1968, James Brown's "Say It Loud—I'm Black and I'm Proud" was a well-known song during protests that praised Blackness as opposed to concealing it in a religious association. As the decade progressed, musicians shifted away from their straightforward uniforms to more experimental fashions. Baldwin's style seemed to follow theirs as he began to abandon his tamer looks from his youth in favor of big frame sunglasses, more extravagant coats, and colorful ascots and scarves exuding the coolness of a Black dandy. Dandyism was initially imposed on Black men in eighteenth-century England, during the Atlantic slave trade when it became trendy for slaveowners to dress their human property in the latest fashions (Miller 2009). The stylishness and flamboyance of Black dandyism were later used to subvert the markers of slavery and continued oppression in society.

In another photo of Baldwin from 1969 (not pictured), he wears large sunglasses, a one-button tweed blazer (a Miles Davis signature, who was a jazz musician), and a patterned ascot. Although his styling is cool and flamboyant, his sunglasses can be read as an object of solitude. They cover much of his face in the image and give off the same essence of "secrets" and "loneliness" that he portrays in *Another Country*. Even his body language in the photo with one arm crossed over himself is guarded against letting a viewer "in."

During the Black Power Movement in the 60s, the "Black is Beautiful" campaign was also gaining traction as a means to dispel the superiority of Eurocentric beauty. The movement encouraged those of African descent to wear their hair in its natural state and display their afros (Matelski 2012). Pan-African styles also started to become popular including colorful dashikis, braided hairstyles, and large ostentatious jewelry reminiscent of African tribal adornment. Although Baldwin never wore the afro, he seemed to have begun to more outwardly adorn his body, too. In the 1969 image (not pictured), he is also wearing rings. Rings, particularly pinky rings, have always been preserved for people of a certain higher social status (Dezentje 2017). For the Black community, donning accessories like rings is a process of beautifying and signifying. Since Baldwin considered himself ugly, choosing to adorn his body with accessories indicates that he was shifting his thoughts or was gaining confidence in his appearance in relation to the Black revolutionary society around him.

## 2.3. *Just Above My Head*

Baldwin's *Just Above My Head* (1979; Baldwin and Pinckney 2015) is narrated by the character Hall Montana and chronicles his memories of his brother Arthur and their friends. After Arthur's death, Hall is remembering stories about their parents, their friends Julia and Jimmy Miller, and others. The book tackles themes that surround the Black church, racial prejudice, success, sexuality, and self-realization. These themes come together from his previous works such as *Go Tell it on the Mountain* and *Another Country*. Hall remembers the stories of his friends and family through the lens of the white-dominated hegemonic society. Much of Hall's remembering depicts the struggles of the people around him not fitting into that dominant society. Interestingly, readers do not find out how Arthur died or get insight into his death until the end of the book in which Baldwin details the death by writing, "He starts down the steps, and the steps rise up, striking him in the chest again, pounding between his shoulder blades, throwing him down on his back, staring down at him from the ceiling, just above [*his*] head" (p. 1037). The book, however, is titled *Just Above My Head* which implies that Baldwin wanted to suggest that the shadow of death is hovering over his own (or Hall's) head too and that, furthermore, it hovers over all of us. Despite the book not thoroughly explaining Arthur's death until the end, readers do learn that he is gay, a gospel singer, and an activist in the freedom movement which connects to Baldwin's own life.

Through the various themes and the impending death in the book, Baldwin hints at the need for grace, compassion, and forgiveness. He writes, "That energy called divine is really human need, translated, and if that God we have created needs patience with us, how much more patience do we need with God!" (p. 815). Since Baldwin had grown distant from the church because of his apprehensiveness of its effectiveness, perhaps by this time he had grown comfortable in mortal forgiveness. The book, although peripheral to religious association, really emphasizes that people should be tolerant of one another's faults and differences, including racial differences. This idea can also be applied to the views on race relations described in the novel and in broader society. This is, of course, connected to the self-realization of the characters and Baldwin himself.

Baldwin also emphasizes compassion in connection to love. He writes, "The first love disappears, but never goes. That ache becomes reconciliation" (p. 589). Baldwin suggests that pain might never go away, but that pain can be accepted and lived with. This quotation can be linked to Baldwin's thoughts on forgiveness and the role of the past in understanding the present as it relates to racial relations. In 1970, James Baldwin and anthropologist Margaret Mead sat down to discuss identity, race, and forgiveness. The transcript became a co-authored book called *A Rap on Race* (1971). Baldwin and Mead discuss forgiveness, which has religious roots, roots that are complex and contradictory, particularly as it relates to slavery (Mead and Baldwin 1971). Baldwin said, in the transcript, "I, at the risk of being entirely romantic, think that is the crime, which is spoken in the Bible, the sin against the Holy Ghost which cannot be forgiven. And if that is true . . . ". Mead cuts him off, "Then we've nowhere to go." Baldwin then responds to her, "No, we have atonement" (Mead and Baldwin 1971). In Christian theology, atonement is considered the reconciliation of God and humankind through Jesus' sacrificial death. It is through compassion and the love of God for humans that forgiveness can be awarded. During this time, it is evident that Baldwin was very interested in the idea of forgiveness through love and compassion, on a human level though, rather than through religion. Although he holds on to features of his Christianity, he may have been using it as a prophetic responsibility to society and thus in a secularized activist context.

In relation to acceptance and self-realization in *Just Above My Head*, Baldwin includes multiple "truths" in Arthur's story. Despite the book beginning with Arthur's death, Hall has to write the memoirs of several other characters to reach that part of the story. Through getting to the "truth," Baldwin is sure to remind readers that *the* truth of which he speaks comes only through intense struggle. For example, readers have to go through Julia's story of incest and rape. Readers also have to journey to the South where Peanut is killed by

white men. Baldwin writes, "Sometimes you hear a person speak the truth and you know that they are speaking the truth. But you also know that they have not heard themselves, do not know what they have said: do not know that they have revealed much more than they have said. This may be why the truth remains, on the whole, so rare" (841). In the book, Baldwin reveals *truth* through the characters, but not *the* truth, or what happened to Arthur. The horrible truths of each character aid readers in understanding the full story. In trying to learn about Arthur's death, readers are in constant combat with forces (the other characters) that deter them from the truth of Arthur's death. At the end of the book though, there seems to be a lesson on the role of truth in healing and self-realization. Baldwin writes, "Oh, my loving brother, when the world's on fire, don't you want God's bosom to be your pillow? And I say to him, in my dream, No, they'll find what's up the road, ain't nothing up the road but us" (p. 1039). This self-realization and "truth" is only revealed through the stories of violence and oppression of the other characters. Therefore, the readers must "earn" the story of Arthur's death, by understanding the stories of everyone around him.

By 1971, Baldwin had relocated to the South of France, but neither his writing style nor his sartorial style had lost its vitality. In *Just Above My Head*, Baldwin exemplified his quest for a fullness in spiritual belief: one that embodied love, compassion, forgiveness, and truth. His fashion choices emulated this essence. In Figure 3, a photo of Baldwin in 1972, he is pictured wearing a snappy black utility jacket, a silk paisley scarf, oversized driving sunglasses, and a large ring. By this time, modern menswear has splintered off in varying, new directions, but Baldwin continued to adorn his body like he pursued writing, sharply and detailed, with dandy tendencies. Baldwin did not waver on the subtle details he is known for stylistically. He seemed to have kept up with his patterns, neckwear, and jewelry. While 70s U.S. fashion became loud and featured flares and bell bottoms, prints, and unabashed color pairings, Parisian fashion during the time was a little more muted. French style included slimmer cuts, smaller waists, and a straighter leg in the pants. Suits were popular for all occasions, but there was popularity in more informal styles. In France, flannel or western shirts, sweaters, jeans, khaki chinos, leather jackets, and oxford shoes were popular for menswear. In Figure 4, a later photo of Baldwin from 1983, he is styled down in a yellow sweater, trousers, and brown boots. His accessories are subtle, but apparent, featuring a silver bracelet, a watch, and a gold and silver ring reminiscent of Black fashion in the States. His body language is relaxed and reclused, with a sense of half intellectual, half musician. He sits atop a rock, in nature, alluding to the fact that he has escaped from his chaotic, urban, setting in the States.

Throughout his career, not only did his writing progress towards his self-actualization, but his sartorial style too indicates that he became more comfortable and enduring in his appearance. Baldwin's writing often hinted at how political and/or social circumstances worked against the individual, making one feel disconnected or not whole. Despite his struggles with the Black church, and at times, Black politics, Baldwin's sartorial aesthetic represents a sense of bodily healing, connected to collective Black narratives.

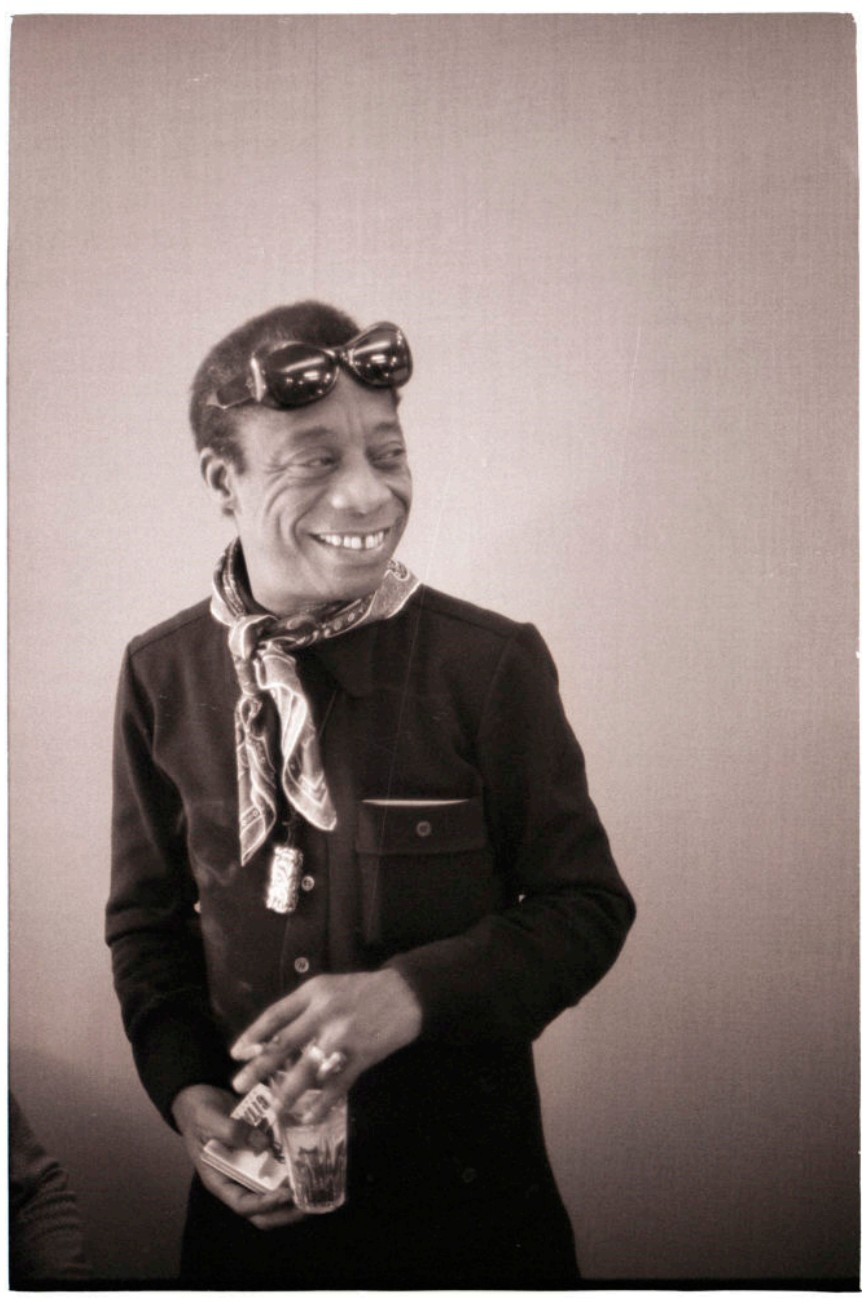

**Figure 3.** Photo by Sophie Bassouls/Sygma via Getty Images.

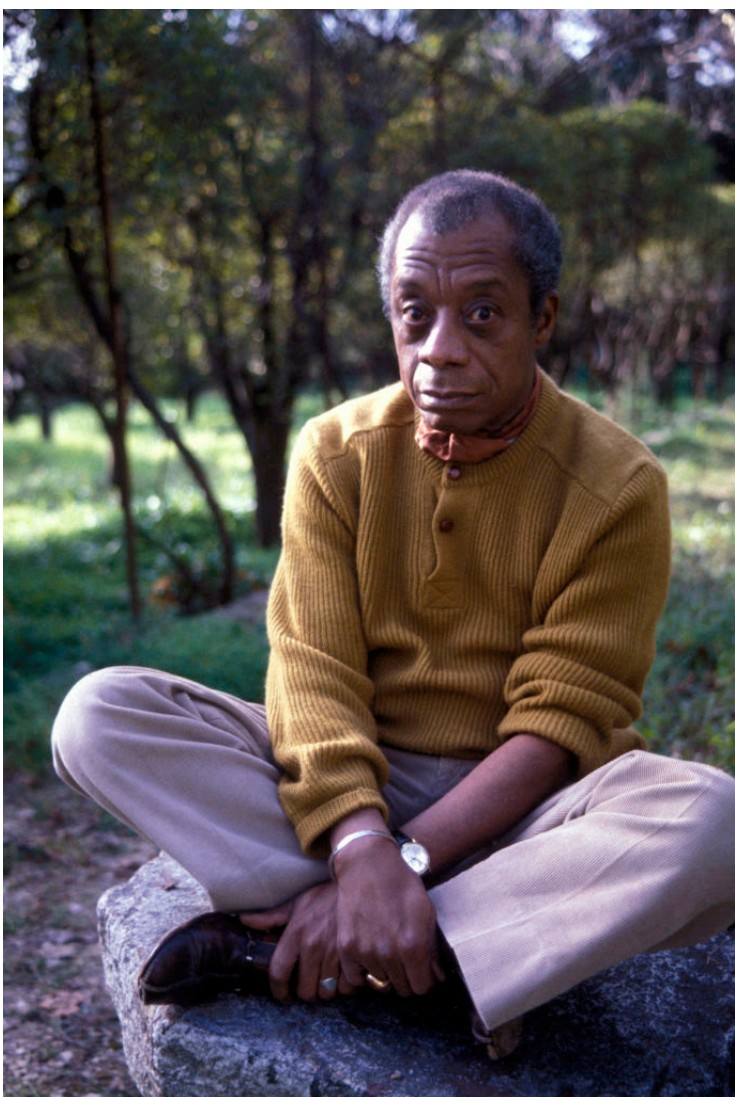

**Figure 4.** Photo by Micheline PELLETIER/Gamma-Rapho via Getty Images.

### 3. Discussion

My hope is that this reading contributes to a more informed analysis of the role that fashion plays in the assessment of a writer's creative and spiritual positionality. While the writer, Baldwin, clearly used his literary style to express himself, a social semiotic analysis suggests that he also used his body as a medium for spiritual and political self-expression. A look at his signature sartorial and literary style using excerpts from his works and archival photography illuminates the many connections between literature, society, and the body. Through this analysis, it is clear that Baldwin's thinking on religion, race, social injustice, America's role in the modern world, *and* his fashion choices remain prescient and timely.

### 4. Conclusions

James Baldwin proved a unique subject to study because of his esteem in current pop culture. His sartorial choices have meaning-making potential that is relevant even today. Through my analysis, it is evident that there are many historical antecedents of clothing inspiration that demonstrate the ways in which fashion responds to the larger world. While Baldwin's writing has itself prompted conversations around the complexities of Black authors, his bodily narratives through sartorial choices, too, instigate an interesting phenomenon. In the contemporary context in which we engage with literature and authors, interacting with literary works, author imagery, and the societies that we are enmeshed in

offers an invigorated analysis. With the advent of social media, incorporating images into narratives is increasingly important to engage viewers and readers. Since the visual has become ubiquitous and part and parcel of our daily lives, it has a tremendous impact on every facet of our personal and professional lives, including how we engage with literature.

The investigation of text and visuals allowed me to explore James Baldwin's profile, his selected prominent works, and signature sartorial moments that expressed his persona. The examination of his clothing and writing, together, also allowed me to explore his historical trajectory and distinctive features of sartorial and writing style. Revealing anecdotes about Baldwin's life in the social context in which he wrote his works also illuminated that the body can be a site for spiritual, political, and historical memories.

This study can and should be expanded in many ways. The present study was limited to only three of Baldwin's works. It is possible to extend the project to his other works drawing on the analytic framework presented. The present study was also limited to making connections between Baldwin's sartorial and literary style to broader Black narratives of style. Another expansion of this work might include an in-depth critical analysis of archival photography of Baldwin and offering innovative readings of his literature based on that analysis. A richer analysis of Baldwin's style might also consider the other cultures in which he navigated and intersections with consumer marketplace and industry expectations. Additionally, analyzing Baldwin's fiction noting dress and style in his characterization, and comparing those descriptions with images of him and his style across time and place would also be a worthwhile study. I recognize the limits of the present study, but offer this initial analysis as a starting place for much more narrating.

**Funding:** This research received no external funding.

**Data Availability Statement:** Archival data available in publicly accessible repositories.

**Conflicts of Interest:** The author declares no conflict of interest.

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
