# Peer review of "The Narrative Bodies of James Baldwin: A Discussion of Literary and Sartorial Style"

_2410-9789, doi:10.3390/literature2040017_

Round 1

Reviewer 1 Report

Thank you for sharing this creative approach to Baldwin studies and for pursuing an under-theorized topic in the field – Baldwin’s thoroughly individual sartorial style. I found your argument that Baldwin’s fashion may serve as a reflection of “the creative and spiritual essence of his work” to inspire curiosity and interest, and I was particularly appreciative of the archival research your essay conveys in the context of photography and portraiture of Baldwin.

            The essay provides a thoughtful overview of iconic photographs of Baldwin and an engaging close reading of the clothing and style captured in these photographs.  The strongest aspects of the essay are the analyses of the photographs themselves.  In order to strengthen the argument and provide a more nuanced reading of the novels chosen as representative of Baldwin’s creative style, I suggest the author deepen their analysis of Baldwin’s style as captured in this photographic record. The article currently begins each section with a summary of each novel and its key themes, and the literary readings themselves often present longstanding interpretations of the texts. To create more of an opportunity for innovation in the actual readings of the literature, I propose that the author foreground the analysis of Baldwin’s sartorial style and then turn to the literary analysis, using insights from the social semiotic analysis as a window into a new way of understanding these important literary works.  As of now, it isn’t clear how the discussion of clothing is actually contributing information or new perspectives on the readings of the texts.

            Further, I would recommend the author consider Baldwin’s clothing and his literature as not only a means of self-expression, but a text that is shaped by a larger social world. The argument would be strengthened, for example, if there were greater historical background and context about the clothing and in particular the context in which the photographs were taken.  What was the occasion for these photographs, and the clothing choices captured within?  Who were the photographers and what was Baldwin’s relationship to them? What was the intended audience of the photograph?  For example, figure 1 is a photograph taken by Carl Van Vechten, a highly influential literary figure in the Harlem Renaissance and also a queer man who would have been part of Baldwin’s queer literary circle in New York in the 1940s. How does the context of this photograph differ, then, from the second photograph taken by Baldwin’s publicist at Knopf? A rich analysis of material culture would consider how Baldwin’s style is shaped by the larger cultures through which he navigated, the consumer marketplace, and the expectations projected onto him as a Black author and public intellectual. 

            I would recommend the author read Magdalena Zabrowska’s Me and My House: James Baldwin’s Last Decade in France (2018) for a rich example of a study of Baldwin focused on material culture, and Monica L. Miller’s Slaves to Fashion: Black Dandyism and the Styling of Black Diasporic Identity (2009). 

Reviewer 2 Report

The author takes sartorial expression and literary style seriously as sites of interrogation. What is particularly fascinating for the reader is to see how Baldwin, in many ways, presaged currently early twenty-first century mediagenic approaches to literature in which readers are more than ever likely to have a sense of what writers of novels look like. The advent of social media has arguably changed how reading publics engage with literature. While paper books have not been completely supplanted by various digital forms as “old” and therefore “obsolete” technology, readers are nevertheless invited to connect with authors through their various social media platforms even from the point of sale, for instance, from larger vendors. What do we make of these cultural moves and how do we begin to unpack their significance for our own time as readers/viewers? In addition to its contributions to analysis of Baldwin as a writer, this article is valuable, methodologically, for its presentation of how to assess the significance of style as both literary and visual phenomena. The author is encouraged (see the specific recommendations, below) to further explicate the significance of these research strategies in the conclusion of the article.

The three works selected, Go Tell It on the Mountain (1953), Another Country (1962) and Just Above My Head (1979) were chosen by the author as representative of different periods in the writer’s life as a sartorial subject and a literary author. While this study is limited to these three works, given the richness of the analysis, might it be possible to extend the project, for another longer publication, in the future such as a book length study? This comment is not meant to take away from the analysis presented. Instead, it is intended to signal that the importance of the work undertaken and to suggest, for a later date, that the author consider a longer study drawing on the analytic framework presented in this article.

Specific Comments for Suggested Substantive Revision

  • The author opens with the provocative statement on page 1, line 21 that “Fashion is a language.” The author then goes on to suggest that fashion is a “wordless means” of communication. While this opening makes for a captivating entry point for bringing visual and literary textual analysis into dialogue with each other about style, it could be strengthened with some reference to some classic works on the concepts of “style” and “stylization” and their links to notions of “race” and “racialization.” See for example the works of Black British sociological and cultural studies scholars such as Carol Tulloch (The Birth of Cool: Style Narratives of the African Diaspora); Stuart Hall in his many explorations of “cultural studies” and what it means to study culture and the links to race and Paul Gilroy (There Ain’t No Black in the Union Jack). In the US context, the works of photography and visual cultural studies scholar Deborah Willis on black bodies and their representation is also significant and recommended. See for example Willis’ Posing Beauty: African American Images from the 1890s to the Present. Also recommended for historical contextualization is Shane White and Graham White’s Stylin’: African American Expressive Culture from the Beginnings to the Zoot Suit.
  • The conclusion which begins on page 12 could be expanded to include some reflections on the significance of the study for research methodology for exploring visual and literary texts and style. The conclusion, as it stands, is a brief reiteration of aims of the article. For example, when the author says on page 12, line 339 about Baldwin that “[h\is sartorial choices have meaning-making potential that are relevant even today” providing some more details here would be helpful for the reader.

Reviewer 3 Report

I do absolutely think that examining Baldwin's style​ and sense of fashion is a worthy one. I don't think the general semiotic frame adds anything here. Better from my perspective would be to really pin certain photos and their characteristics as precisely as possible to time / place and in relation to things in JB's novels and essays from the period. Also, I'm not sure how fashion and religion come together given the magazine to which this essay has been submitted. Below I'll list page-by-page a few thoughts. 

page 2 section 2. line 2: I don't think it's accurate to say that JB's first novel "propelled him into literary stardom. . .". Baldwin's road to fame was somewhat more gradual. By the late 50s one could say that he was a "successful writer" in that he'd become well-enough paid to live from his art beyond questions of immediate subsistence. But I don't think it's true that he was "famous" until late 1962 when the essay "Letter From a Region in My Mind" appeared in the New Yorker and then became The Fire Next Time in early 1963. So 1963, after three novels and two volumes of essays, with his sixth book, I think it's accurate to say that Baldwin

had achieved "literary stardom." 

Generally, I don't see what Biblical allusions in Go Tell It On the Mountain have to do with Baldwin's style of dress. I would think looking closely in the text of the book for descriptions of characters' dress would be worthwhile in Go Tell It.... as well as the later novels touched upon in the essay. 

page 3: second graph. . . Baldwin was broke during the years just before and just after Go Tell It... was published. In letters he continually asks family for money, lives in borrowed spaces, etc. He had to borrow suits for photographs and complained that he had no money to get his own clothes out of the cleaners. So I'd say a precise discussion of JB's sartorial choices would have 

to come to terms with the fact that he was broke at the time. That would change by 1955 or 56. 

Also graph two: the note about denim really belongs in the section devoted to Another Country, 1962, wherein Baldwin discusses the appearance of blue jeans and denim and the physiques emphasized by the changes in dress.    

graph 3: I wouldn't say a look of "casual slacks" and "a relaxed and effortless look" matches Baldwin's rather desperate and struggling life at this early stage in his career not the intense and at times tortured tone of this fiction in Go Tell It. . . and Giovanni's Room. One could, I think, detail a certain self-conscious elegance in the language, but its author would have 

a hard time dressing the part of a "Jamesian" [that's Henry James] that his prose positioned him to be. 

A loosely ironic or angular relationship between appearance and reality, therefore, seems to be a signal of Baldwin's style both in artistic and sartorial terms. 

page 4: graph one: I don't think it's accurate to conclude that Baldwin considered the church "a safe place."  

graph 2: I wouldn't say "Baldwin eventually left the church." It was rather abrupt, his exit, and tortured, when he as 17and a high school friend helped him free himself from his role in the churches of Harlem. That was in 1941. Baldwin wouldn't stand in the pulpit of a church again to speak until May 1960. 

graph 2: Here's where we can say Baldwin becomes a literary star. 

page 5: I think it's worthwhile to note that William Cole was JB's agent at the time of this photo. I wonder about how the mustache (which Baldwin never wore again) relates to his masquerade as a masculine and heterosexual author at the time. Overall I find a fluid and changing sexual significance in Baldwin's evolving style. After the mid-60s he strayed away from the solidly 

"masculine" earlier styles adopting rings and scarves, etc. 

The author here again comments upon JB's "more casual and relaxed" but in Fig. 2 he looks pretty stiff, leaning back with brows down. True no tie but still. In this paragraph again the author writes that Baldwin "was beginning to shy away from and question mainstream religion." Baldwin was a Trotskyite during WWII and hadn't been "out of the church" for over a decade by the early 1950s. 

The final sentence about Black society's "rejection of the middle-class, Christian, properly well-dressed. . ." doesn't seem to fit the period either. Certainly the protest traditions in the early 60 as well as daily life of many upheld those appearances. 

Section 2.2

page 5: I don't think it's accurate to say that Another Country "is about the life of Rufus Scott." The narrative moves between several important characters. Maybe the novel centers upon the death of Rufus Scott. But I think it would be better to characterize the book as about

a community of diverse origins such as Baldwin signals in the epigraph from Henry James. . . and "unprecedented multitude." 

page 6 graph 2: this is one of the stronger graphs I think because it's based upon quotes form the novel about loneliness instead of general claims about themes. 

graph 4 seems the author is drawing from the essay "The Uses of the Blues" and it would be better to quote Baldwin's language there. Later in this paragraph historical periods get out of sync with discussion of the Black Power Movement and Another Country / early 60s. For Baldwin's fiction in the Black Power Movement I'd go to Tell Me How Long the Train's Been Gone (1968), especially the "Black Christopher" section at the end. 

I wouldn't say that the Black Power Movement was the first era in which "secular music started to influence the styling of Black bodies." One thinks Harlem Renaissance “jazz age” styles and, famously, of Bebop stylings with high drape pants (as in Billie Holiday’s “Fine and Mellow” as well as many photos) and berets and shades, etc. And also the late-Swing era ballroom culture with Zoot Suits in the early 40s. I'd bet there were aspects of Blues styles in juke joints and rural hole-in-the-wall clubs as well. 

I think I know the photo (from Life magazine, May 24, 1963) with Baldwin and Meredith. But it doesn't appear in the essay that I have. Also the coupling in the paragraph between Meredith from 63 and "Say It Loud. . ." in 1968 seems to confuse eras again. 

The point about Baldwin's shift in dress in the late 60s is accurate. Notably he spent a lot of time in California between spring 1967 and summer of 1969. Also accurate that a similar shift in Miles's Davis work / dress happens around that time. Baldwin and Davis were well acquainted and almost exactly the same age. So both were in the mid-late 40s during this era. 

Baldwin has a famously and complexly contested relationship to some of the Black revolutionaries of the era. The question of his sexuality was a subject of intense criticism and ridicule. I think there's a lot of interesting work to be done concerning how Baldwin's more openly queer styles at the time related to the hyper-masculine styles of many of the political revolutionaries as well as the "Afro-centric" styles of the Cultural Nationalists, which two wings of Black Power were themselves (at times violently) at odds. 

Page 7: I'm not so sure I'd read JB's sunglasses as "solitude." See Robert Farris Thomson's "Aesthetics of the Cool" essay from 1973. Also of note re: the early 60s would be Baldwin's discussion of "Black style" versus "white style" and "make believe" from the film Take This Hammer. See A Dialogue, Baldwin's book with Nikki Giovanni for his apprehensions about "Black is Beautiful" and slogans in general. 

bottom of page 7: again, Baldwin styles in the late 60s and early 70s have a complex relationship to "Black revolutionary society around him," and by then he's a generation older than the activists.... Baldwin never picked his hair out into an afro, etc. 

page 8

graph 1: I think the discussion of themes in Just Above My Head is rather too general. It's absolutely not true that Hall "remembers the stories of his friends and family through the lens of the white-dominated hegemonic society." One of the major vulnerabilities of JB's narration in his final novel (as well as the previous one) is how an "every day" kind of man like Hall (or Tish in BEALE ST) come up with radically and complex dissents from various powerful hegemonies. The answer is that Baldwin eschewed that particular constraint of first-person narration in realistic fiction by then. The narrator, in fact, is Baldwin and he makes little effort to disguise it. At the end of this graph I think it's crucial that Arthur is a gospel singer and activist / publicist in the freedom movement, which echoes Baldwin's role as a politically engaged artist during those years. 

graph 2: the question of racial tolerance in Baldwin's fiction generally, and especially his later fiction, is complex. His explicit and implicit critiques of white characters in Just Above My Head, especially one named Faulkner is rather more withering than tolerant. 

Bottom page 8: the question of atonement as relates white people, but the late 1970s, for Baldwin, had to do with material action in atonement for white atrocities and assumptions. This, Baldwin argued, might or might not help​ Black people, which held Black people, thought Baldwin, might or might not need if it existed at all. 

But, crucially, Baldwin thought, such material atonement (meaning: change the way you live) was the only hope while people had to regain a grasp of functional reality in and around their lives. Baldwin thought that such action on the part of people "who think they're white" was one of the only--maybe the only--ways white people could regain a coherent relationship to history, to the real world. 

page 9: The stakes, generally, in Just Above My Head, and really long since in Baldwin's fiction, are complexly collective and communal. "Self-realization" indicates a far more individuated process than Baldwin believed in by then. Even in Go Tell It. . ., often read as the process of John's "self-realization," the stakes are far more than individual, far more like those of Joyce's character in Portrait of the Artist. . . for example. By Just Above My Head, in these terms, Baldwin had gone far far beyond where he'd begun in his first novel.   

bottom of page 9: Looks to me as if that photo (figure 4, sitting on the rock) was taken in Baldwin's yard in St. Paul de Vence. There was a path through fruit trees that led to the road along the side of the property; it looks like they're on that path. 

One final note is that the figures noted in the text don't always match those in the essay. Obviously a strict correspondence between the figures analyzed and the ones noted is necessary in an essay like this. 

My recommendation would be to work through Baldwin's fiction noting dress and style in his characterization and associate those descriptions with images of him and his style from the periods (and maybe places) that correspond to fictive settings. As is in this essay the connections tend to be too general and, at times, historical eras become confused. 
